# Polyethylene Microplastic Particles Alter the Nature, Bacterial Community and Metabolite Profile of Reed Rhizosphere Soils

**Zeyuan Tian [1], Biao Liu [2], Wenjun Zhang [2], Fan Liang [2], Junfeng Wu [2], Zhongxian Song [2] and Yichun Zhu [1],***

[1]  School of Resources and Environmental Engineering, Jiangxi University of Science and Technology, Ganzhou 341000, China
[2]  Henan Key Laboratory of Water Pollution Control and Rehabilitation Technology, Henan University of Urban Construction, Pingdingshan 467036, China
***  Correspondence: zhuych@jxust.edu.cn

**Abstract:** With the wide use of polyethylene film, the influence of polyethylene microplastic particles produced by its weathering on the rhizosphere soil microenvironment has attracted more and more attention from scientific research circles. In this study, the effects of low (0.2% $w/w$), medium (1% $w/w$), and high (2% $w/w$) doses of polyethylene particles and the combined reed biomass (2% $w/w$) on soil environmental factors and bacterial communities and metabolites in the reed rhizosphere were evaluated by a 90-day pot microscopic simulation system. The shape and surface microstructure of polyethylene particles in each treatment group changed obviously. A high (2% $w/w$) dose of microplastics significantly increased the TKN, TOC, and TP in reed root soil. The addition of the biomass significantly improved the activities of urease and sucrase in the soil. The α diversity of bacteria was not significantly affected by the addition of LDPE microplastics and biomass, but the β diversity of the bacterial community and the relative abundance of the Candidatus_Roku Bacteria, Chloroflexi, Unclassified_Blastocatella_Genus were significantly changed by the addition of middle (1% $w/w$) and high (2% $w/w$) doses of microplastics. In addition, the spectrum analysis of the soil metabolites showed that the abundance of soil metabolites was changed in each treatment group, and the differential metabolites were significantly up-regulated or down-regulated. Our findings provide a scientific reference to elucidate the impact of LDPE microplastic particles on the inter-rooted soil microenvironment and improve our understanding of the potential risks of microplastics in soil ecosystems.

**Keywords:** polyethylene microplastic particles; biomass; rhizosphere soil; enzyme activity; bacterial diversity; metabolites

## 1. Introduction

In recent years, microplastic pollution in ecosystems has attracted widespread global attention. As a new persistent pollutant, microplastics (MPs) were first discovered in the research on marine plastic debris in the 1970s [1]. As the issue of microplastics in the marine environment has been studied in recent decades, researchers have found that the impact of microplastic particles on the Earth's environment has been grossly underestimated. Compared to marine ecosystems, terrestrial soils contain 4–23 times more microplastics than those found in the ocean [2]. In addition, a growing body of research suggests that microplastics may pose a serious threat to the vital functions of terrestrial ecosystems [3,4]. Recent research shows that microplastics are not only ubiquitous in soil but can also last for a long time and are difficult to degrade, which directly affects soil properties, soil substance and its nutrient cycle, microorganisms, and plants [5,6]. Most microplastic particles can reduce the structural integrity of soil, resulting in a decrease in its water-holding capacity. Yu et al. [7] found that different kinds of microplastics have fluctuating effects on various soil enzyme activities. PE microplastics with different molecular weights have a significant

effect on the soil–microorganism–plant. For example, PE particles increase the number and α diversity of rhizosphere microorganisms but inhibit the biomass and photosynthetic rate of plants and destroy the normal mineral nutrition metabolism [8]. Falsini et al.'s [9] research showed that microplastics, such as PE, had negative effects on plant growth in the early stage of exposure, including a decrease in photosynthetic activity and a change in the plant concentration of macronutrients and micronutrients. In addition, microplastics can absorb other pollutants (such as heavy metals and antibiotics) in the soil, which has a far-reaching impact on the ecosystem [10]. Therefore, it is very important to study the risk assessment and degradation of microplastics in terrestrial soil.

Low-density polyethylene (LDPE) is the main material used for agricultural plastic films [11]. In recent years, LDPE agricultural film has become a major terrestrial ecological pollutant due to the widespread use of on-farm mulching techniques. After these films enter the soil, they can be weathered and split in a short time, eventually forming microplastic particles [12]. In general, it is considered that LDPE is easy to decompose and has a low impact on the soil environment, which is why it is widely used based on these advantages. However, existing research shows that microplastics formed by the decomposition of LDPE are highly hydrophobic and difficult to degrade, which makes them easily affected by physics, chemistry, or biology [13,14]. The long-term retention of LDPE microplastic particles in the soil has a certain effect on the soil ecology, but its mechanism is still controversial and needs further study.

In recent years, research on the impact of microplastic particles on soil ecology has become a hot topic, but there are fewer relevant reports on LDPE. Qi et al.'s [15] research showed that the addition of 1% LDPE microplastics significantly affected the bacterial community composition in wheat rhizosphere soil within 4 months and increased the relative abundance of specific taxa. Fei et al. [7] found that both polyethylene and polyvinyl chloride microplastics reduced the bacterial abundance and community diversity and significantly altered the bacterial community structure, stimulating the growth of nitrogen-fixing bacteria while suppressing the relative abundance of *Sphingomonas* and *Flavobacterium*. In addition, microplastics can change the metabolites of rhizosphere soil (such as root exudates and microbial metabolites) to promote the formation of new dominant microbial communities, which further reveals the influence mechanism of microplastic residues on the soil microenvironment [16]. Previous studies have shown that nano-materials entering the soil changed the spectrum of the metabolites in the soil and significantly changed the related metabolic pathways, such as sugar and amino acids [17]. However, it has been shown that basic metabolic pathways, such as carbohydrate metabolism, lipid metabolism, and nucleotide metabolism, are affected by large amounts of microplastic residues [18]. However, research on the effects of microplastics on soil microbial metabolites and metabolic pathways and functions is still lacking. More research is needed to determine which soil metabolites and metabolic pathways can be affected by microplastic particles and to elucidate the impact processes.

Therefore, this study combined macrogenomic sequencing and untargeted metabolomics based on a macrogenomic technology and the metabolomics platform to reveal the effects of LDPE microplastic particles on rhizosphere soil's physicochemical properties, enzyme activity, bacterial communities, and metabolites. These results will help to enhance our understanding of the potential risks of microplastics to soil ecosystems.

## 2. Materials and Methods

### 2.1. Experimental Materials

For the accuracy of the experiment, the topsoil (0–20 cm) of sediments along the Li River, which is far away from farmland and has less human activity, was selected. We air-dried indoors, removed impurities, sieved with a 2 mm mesh, mixed thoroughly, and stored for later use. The initial physical and chemical parameters were, determined as shown in Table S1.

LDPE particles (low-density polyethylene and chemically pure) were purchased from the UBE-Japan Ube Yuko Co., Ltd. The LDPE microplastics had a particle size of 150–180 μm. The pyrolytic properties were determined using an SDT-Q600 thermogravimetric analyzer in $N_2$ gas at a heating rate of 10 °C/min in the range of 25–800 °C.

The reed (*phragmites australis (cav.) trin. ex steud.*) biomass was obtained from the Li River wetland in Pingdingshan, Henan Province, China. Fresh reeds were brought back to the laboratory and rinsed with sterile water, cut into small pieces, and deactivated at 105 °C for 20 min, then quickly cooled to 65 °C for 24 h, and then removed and sieved to a diameter. The reed seedlings were cultivated artificially.

## 2.2. Experimental Description

In this experiment, 24 root boxes were set up, and 9 kg of soil samples were added to each root box. According to the experimental requirements, LDPE microplastics were added according to the mass ratio (microplastics/soil), and four concentration gradients were set in microplastics, which were 0% (*w/w*), 0.2% (*w/w*), 1% (*w/w*), and 2% (*w/w*), respectively. The root box was further divided into two groups, A and B, in which group A did not add biomass, and group B added biomass, according to 2% of the soil quality. There were four treatments (different microplastic concentrations) in group A and group B, and each treatment had three parallel treatments. The different experimental groups and treatment marks are shown in Table 1.

**Table 1.** Name of treatment group.

| LDPE Microplastic Dose | Reed Biomass Dose | 0% (Group A) | 2% (Group B) |
|---|---|---|---|
| 0% | | 0 L | 0 wL |
| 0.2% | | 0.2 L | 0.2 wL |
| 1% | | 1 L | 1 wL |
| 2% | | 2 L | 2 wL |

Notes: Where 0, 0.2, 1, and 2 represent the amount of microplastics (0: 0%, 0.2: 0.2%, 1:1%, and 2:2% *w/w*), and w and L denote the addition of the reed biomass and the planting of reeds, respectively. There were three parallel samples for each treatment group.

At the beginning of the experiment, three reeds were planted in each root box. After seven days, only one reed with a similar growth condition was kept in each root box. The soil moisture content was kept at 60% by the weighing method, and it was cultivated at room temperature and in natural light. After the end of the experiment (the 90th day), soil in the reed root was collected, one part of which was stored at −40 °C for metagenome sequencing or high-throughput sequencing and metabonomics analysis, and the other part was used for physical and chemical parameter determination and microplastic collection. Three biological replicates were carried out for all sample parameters in this study.

## 2.3. Observation of Surface Morphology Changes of LDPE Microplastics

LDPE microplastic particles were collected from the soil in each treatment group after 90 d. The samples were dried at 60 °C for 24 h and used for scanning electron microscope (SEM) analysis and energy dispersive spectrometry (EDS) analysis (Zeiss Gemini Sigma 300 SEM, Zeiss, Jena, Germany). The initial LDPE microplastic was used as a control.

## 2.4. Determination of the Physicochemical Properties of Reed Rhizosphere Soil in Each Treatment Group

The ammonia nitrogen ($NH_4^+$-N), nitrate ($NO_3^-$-N), nitrite ($NO_2^-$-N), total phosphorus (TP), and total organic carbon (TOC) of the soil with different amounts of primary microplastics added were determined according to standard procedures. The TKN in the soil was determined using a Kjeldahl apparatus (KjeFlex K360, Lausanne, Switzerland) after sulfuric acid digestion. The pH was measured with a pH meter (Hanna HI98130, Naples, Italy).

## 2.5. Determination of Reed Rhizosphere Soil Enzyme Activity in Each Treatment Group

The collected rhizosphere soil was naturally air-dried and stored at 4 °C for soil enzyme activity analysis. Soil sucrase (S-SC) was determined by 3,5-dinitrosalicylic acid spectrophotometry, and its enzyme activity unit was defined as 1 mg of the reducing sugar produced by 1 g of soil after 24 h. Soil urease (S-UE) was determined by indophenol blue spectrophotometry, and its enzyme activity unit was defined as 1 μg of $NH_3^-$-N produced in 1 g of soil after 24 h. Soil catalase (S-CAT) was determined by potassium permanganate titration, and its enzyme activity unit was defined as the degradation of 1 mmol of $H_2O_2$ catalyzed by air drying in 1g of soil after 24 h. Soil cellulase (S-CL) was determined using the nitrosalicylic acid spectrophotometry method, where 1 unit of the enzyme activity was defined as 1 mg of glucose produced in 1 g of soil after 24 h. All soil enzyme activity measurements in this study used a soil enzyme kit from Biotech Bioengineering (Shanghai, China) Co. Three biological replicates were performed for each treatment group.

## 2.6. Analysis of Microbial Diversity in Reed Rhizosphere Soil

For each treatment group, 1 g of rhizosphere soil was placed in sterile freeze-dried tubes and sent to Biotech Bioengineering (Shanghai, China) Co. Ltd. for DNA extraction and macro-genome sequencing. Genomic DNA was extracted from the rhizomicrobiome present in 1.0 g of soil using an E.Z.N.A™ Mag-Bind Soil DNA kit (OMEGA, Norcross, Georgia, Norcross, GA, USA), according to the manufacturer's instructions. After DNA extraction, DNA was tested for intactness using 1% agarose gel electrophoresis, its purity using NanoDrop2000, and its concentration using TBS-380. The DNA was fragmented by Covaris M220 (Gene Company, Shanghai, China), and the DNA fragment, with about a 500 bp interruption, was screened. The PE library was constructed by using a NEXTFLEX Rapid DNA-Seq kit. After bridge PCR amplification, the Illumina NovaSeq PE250 platform was used for sequencing. The analysis of raw read files of bacterial sequences was consistent with that of the study by Qi et al. (2020) [15].

## 2.7. Analysis of Metabolic Profiling in Reed Rhizosphere Soil

For each treatment group, 15 g of rhizosphere soil was placed in lyophilization tubes and sent to Biotech Bioengineering (Shanghai, China) Co. Ltd. for extraction and determination of soil metabolites. Then, 100 mg of the sample was weighed into an EP tube, and 1000 μL of an extract solution (methanol: water = 3: 1, with an isotopically labelled internal standard mixture) was added. Then, the samples were homogenized at 35 Hz for 4 min and sonicated for 5 min in an ice water bath. The homogenization and sonication cycles were repeated 3 times. Then, the samples were incubated for 1 h at −40 °C and centrifuged at 12,000 rpm (RCF = 13,800× $g$), R = 8.6 cm) for 15 min at 4 °C. The resulting supernatant was transferred to a fresh glass vial for analysis. The quality control (QC) sample was prepared by mixing an equal aliquot of the supernatants from all of the samples.

LC-MS analyses were performed using a UHPLC system (Vanquish, Thermo Fisher Scientific) with a UPLC BEH Amide column (2.1 mm × 100 mm, 1.7 μm) coupled to an Orbitrap Exploris 120 mass spectrometer (Orbitrap MS, Thermo, Waltham, MA, USA). The mobile phase consisted of 25 mmol/L ammonium acetate and 25 ammonia hydroxide in water (pH = 9.75) (A) and acetonitrile (B). The auto-sampler temperature was 4 °C, and the injection volume was 2 μL. The Orbitrap Exploris 120 mass spectrometer was used for its ability to acquire MS spectra in information-dependent acquisition (IDA) mode in the control of the acquisition software (Xcalibur, Thermo, Waltham, MA, USA). In this mode, the acquisition software continuously evaluates the full scan MS spectrum. The ESI source conditions were set as follows: sheath gas flow rate as 50 Arb, aux gas flow rate as 15 Arb, capillary temperature at 320 °C, full MS resolution as 60,000, MS/MS resolution as 15,000, collision energy as 10/30/60 in NCE mode, and spray voltage as 3.8 kV (positive) or −3.4 kV (negative), respectively.

### 2.8. Data Statistics and Analysis

Significant differences in parameters between the different treatment groups were analyzed by a one-way ANOVA test using SPSS 25.0. The shared characteristics between the bacterial species levels of the different treatment groups are shown by Venn diagrams. Differences in bacterial community diversity and metabolites between the different treatment groups were analysed separately using principal component analysis (PCA). Correlations between bacterial communities and physicochemical parameters in the different treatment groups were analysed by redundancy analysis (RDA). In addition, correlations between each physicochemical parameter and species diversity, dominant phyla, and metabolites were analysed using a mantel test. The online analysis platform (Biozeron Cloud Platform, http://www.cloud.biomicroclass.com/CloudPlatform) was used to draw the statistical charts (heatmap, PCA, and Venn diagrams) used in this study.

## 3. Results

### 3.1. Effects of LDPE Microplastic Addition and Reed Biomass on Soil Physical and Chemical Properties in Rhizosphere

In group A, the TKN and TOC increased with the increase of microplastics, while the $NH_4^+$-N, $NO_2^-$-N, and pH were the opposite (Figure 1). With the increase of the microplastic content, $NO_3^-$-N decreased first and then increased, while the TP increased first and then decreased with the increase of the microplastic content. Compared with group B, $NH_4^+$-N, $NO_3^-$-N increased initially, followed by a decrease and then an increase again with the increasing microplastic amount, while the TKN and TP increased and then decreased with the increase of the microplastics. It is worth noting that the TOC content in the reed root soil increased significantly with the increase of microplastics in both group A and group B. In addition, whether in group A or group B, the addition of different LDPE particles had no significant effect on the soil pH value but decreased with the increase of the particles' addition. This indicates that different amounts of microplastics can produce a certain dose-effect on rhizosphere soil.

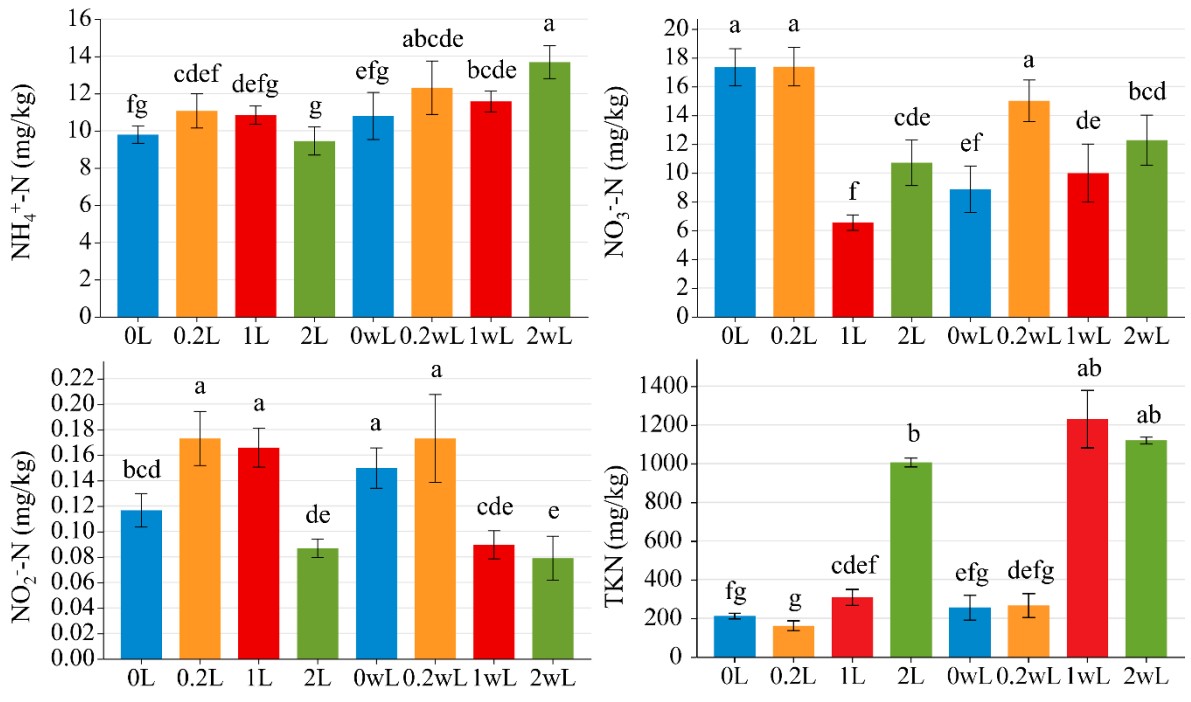

**Figure 1.** *Cont.*

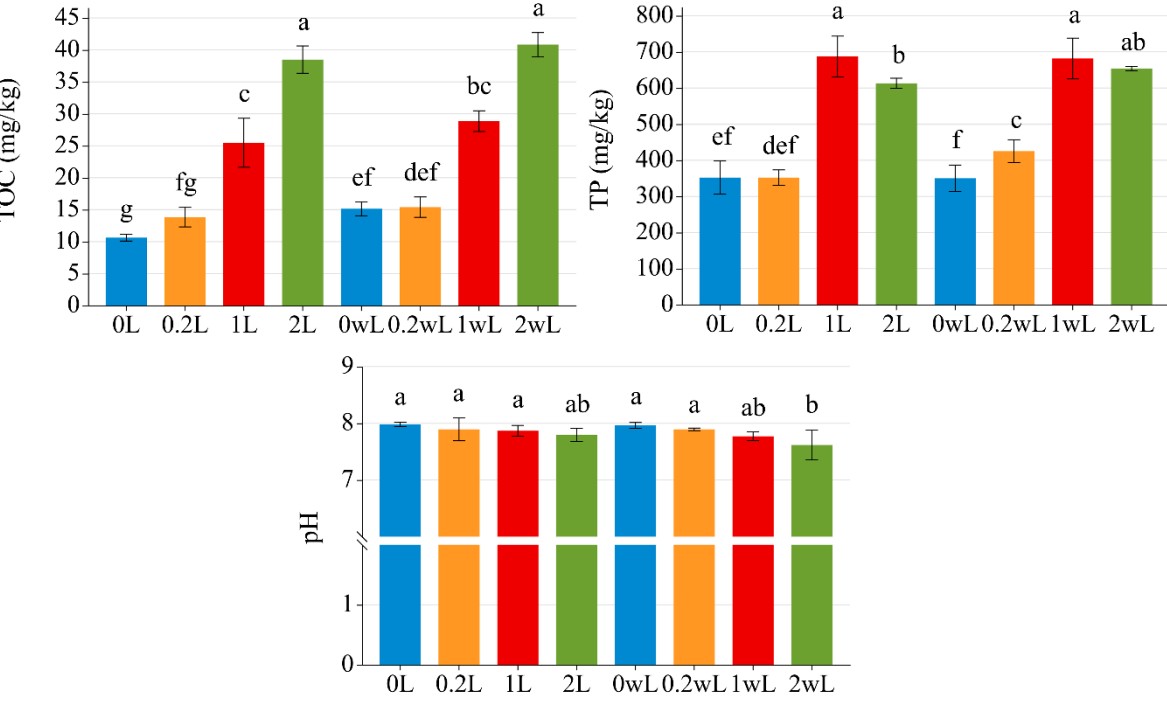

**Figure 1.** Soil physicochemical parameters for each treatment group. Different lowercase (a, b, c, d, e, f, g) letters indicate significant differences between different treatments ($p < 0.05$, $n = 3$). Error bars represent the standard error.

### 3.2. Surface Morphology Changes of LDPE Microplastics

The scanning electron microscope (SEM) micrographs of LDPE microplastics in the initial and each treatment group are shown in Figure 2. The SEM micrograph shows that the surface of the initial LDPE microplastics was smooth and had no depressions, while the surface of the LDPE microplastics cultivated in the soil for 3 months in all of the treatment groups became rough and showed holes, cracks, and irregular deep pits, namely, the surface morphology changed significantly. This indicates that the LDPE microplastics buried in the soil might be biodegraded under the action of microorganisms.

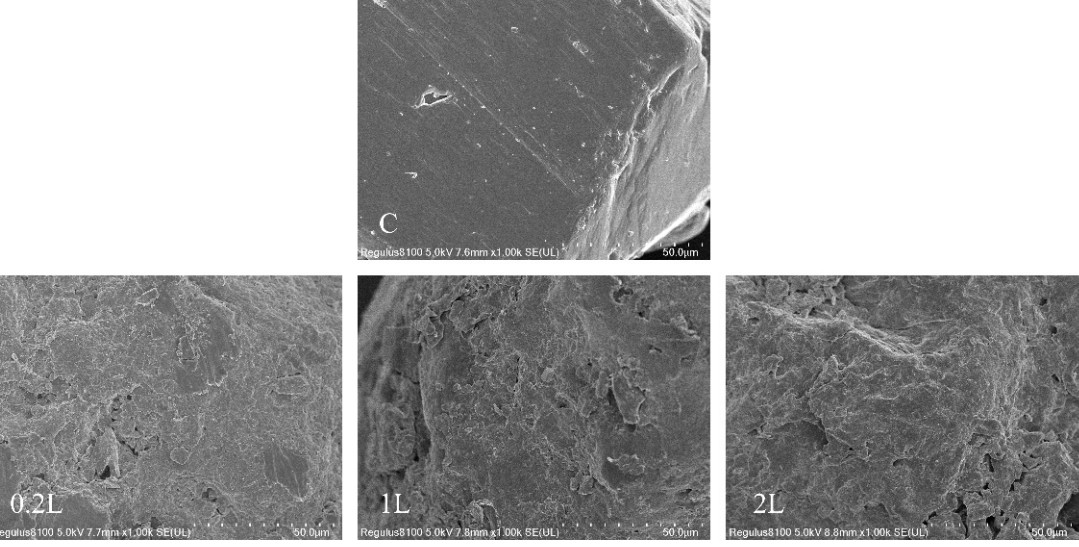

**Figure 2.** *Cont*.

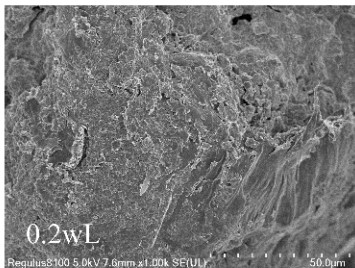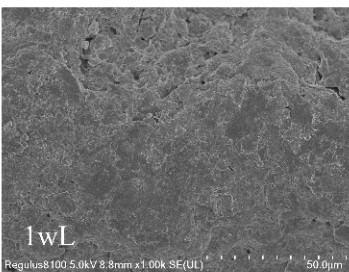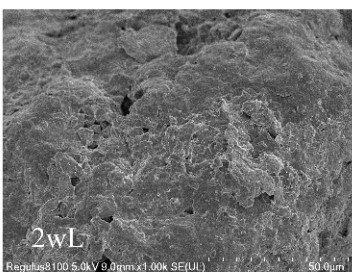

**Figure 2.** Scanning electron microscopy of LDPE microplastic pellets. Raw LDPE microplastic pellets (C).

### 3.3. Distribution of Soil Elements on the Surface of LDPE Microplastics

Energy spectrum analysis showed that the surface element of the initial LDPE microplastics used in this study was mainly C, O, and Na, with atomic percentages of 28.32%, 36.73%, and 34.96%, respectively (Figure S2). After the pot experiment, the LDPE microplastics were isolated from the rhizosphere soil for a surface element analysis. The results showed that the surface of the LDPE microplastics adsorbed a variety of mineral elements and heavy metals. The surface C content of the microplastics increased with the increase of microplastic dosage, which was the same in both groups of the experiments. However, the addition of biomass reduced the content of C on the surface of the microplastics, while O was the opposite. This suggests that the higher amount of microplastics weakened their degradation process, while the addition of biomass contributed to the degradation of the microplastics. On the whole, more substances were adsorbed on the surface of the microplastics in group B. This indicated that the addition of biomass promotes the degradation of microplastics, making their surface rougher and possessing more pore structures, which improves their adsorption capacity.

### 3.4. The Effects of Microplastics on Soil Enzyme Activity

After 90 days of incubation, the effect of the LDPE microplastic addition on the catalase, urease, cellulase, and sucrase activities in the reed rhizosphere soil was measured, as shown in Figure 3. In group A, the addition of 0.2% promoted the activities of four soil enzymes; the activity of soil sucrase increased especially significantly. The activities of the soil catalase, cellulase, and sucrase decreased at first and then increased significantly with the increase of the microplastic dosage, while the activities of the soil urease decreased gradually with the increase of the microplastic dosage. In group B, the addition of the biomass resulted in a significant increase in soil urease and sucrase activities compared to group A. The soil catalase activity decreased first and then increased with the increase of the microplastic dosage, while the cellulase showed a trend of decreasing first and then increasing and finally decreasing.

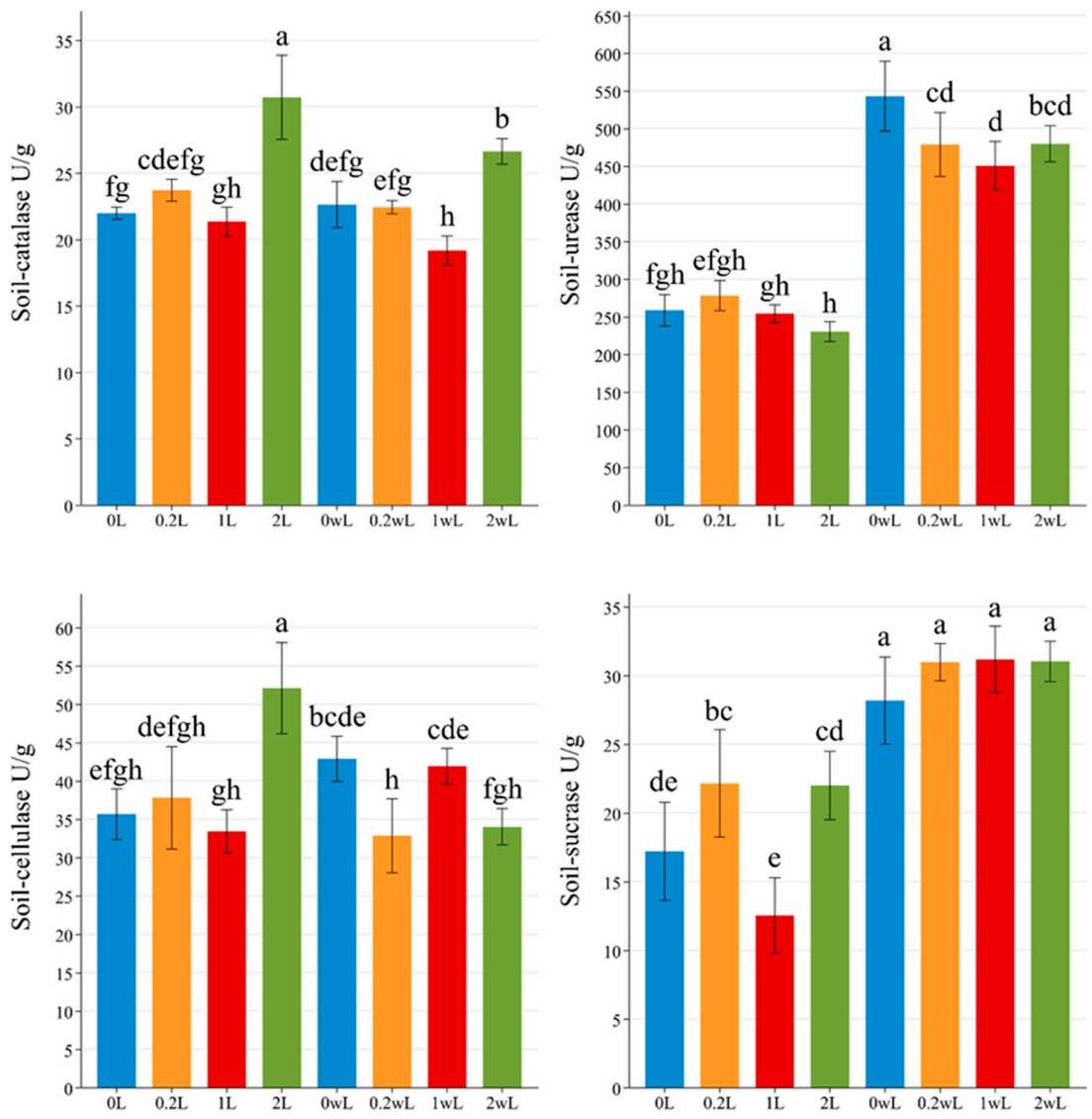

**Figure 3.** Soil enzyme activity in the different treatment groups. Different lowercase (a, b, c, d, e, f, g, h) letters indicate significant differences between different treatments ($p < 0.05$, $n = 3$). Error bars represent the standard error.

### 3.5. Effect of LDPE Microplastics on the Rhizosphere Soil Microbial Community

As shown in Figure S3, the number of species in each treatment group decreased with increasing amounts of microplastic additives, regardless of whether the biomass was added. The addition of the biomass increased the number of species-level microorganisms in the low-dose microplastic addition group compared to group A. The changes in the ACE and observed index that was calculated based on the number of genes also supported the above results (Figure S4). As shown in Figure S3, in group A, the number of species showed a decreasing trend with increasing doses of microplastics. The change in the number of species in group B was different, as the number of species increased in all three treatments (0.2 wL, 1 wL, and 2 wL) compared to 0 wL, but again showed a decreasing trend with an increasing dose of microplastics. The addition of biomass reduced the Pielou index of the microorganisms in the soil, except for the 1 wL. The diversity of the rhizosphere soil microorganisms decreased with the increase of microplastic quantity in group A and showed a trend of increasing and then decreasing diversity after the addition of the biomass.

This suggests that LDPE particles and biomass in the soil altered the homogeneity and diversity of the species in the rhizosphere microbial community.

The soil bacteria at the top 10 phylum level of relative abundance and top 15 genus level of relative abundance are shown in Figure 4a and Figure 4b, respectively. In particular, both a 1% and 2% microplastic addition significantly reduced the relative abundance of *Proteobacteria* in both groups A and B compared to the no-microplastic-addition group, while the opposite was true for *Chloroflexi*. In addition, a 2% microplastic addition significantly increased the relative abundance of *Acidobacteria* (Table S2). The addition of the biomass significantly reduced the relative abundance of *Candidatus_Rokubacteria* compared to the addition of LDPE microplastics-only (Table S2). At the genus level, a 1% and 2% addition significantly increased the relative abundance of the *unclassified_Blastocatellia_genus* in group A compared to the no-microplastic-addition group, while the relative abundance of *Rhodospirillum* significantly decreased with the increasing microplastic dose. The relative abundance of the *unclassified_Blastocatellia_genus* in group B increased significantly at the 1% and 2% microplastic addition levels, while the relative abundance of *Massilia* changed in the opposite direction (Table S3). In addition, the addition of the biomass significantly decreased the relative abundance of the *unclassified_Deltaproteobacteria_genus*.

Significant differences in the bacterial community structure were revealed by a principal coordinates analysis (PCoA), according to the Bray–Curtis dissimilarity at the gene level. As shown in Figure S5, group A showed a clear separation from group B. Among them, 0 L and 0.2 L aggregated with each other, and 1 L and 2 L aggregated with each other in group A. However, 0 L and 0.2 L were dispersed from each other, with 1 L and 2 L. These results show that the 0.2% microplastic addition did not significantly alter the rhizosphere bacterial community of the reed, while the 1% and 2% additive amounts produced significant changes in the rhizosphere bacterial community. The addition of the biomass resulted in varying degrees of overlap between 0wL, 0.2 wL, and 2 wL, but all were clearly separated from 1 wL. On the whole, the reed rhizosphere bacterial community was affected by a combination of the amount of LDPE microplastics and biomass.

The RDA analysis for groups A and B were indicated in Figure S6a and Figure S6b respectively. As shown in Figure S6a, S-CAT (catalase), TOC, TKN, TP, S-CL (cellulase) were positively correlated with 1 L and 2 L, while pH, $NO_3^--N$, S-UE, $NO_2^--N$, S-SC (sucrase) were positively correlated. Among them, TOC ($r^2 = 0.51889$, $p = 0.045$), $NO_3^--N$ ($r^2 = 0.70889$, $p = 0.007$), and TP ($r^2 = 0.62499$, $p = 0.024$) were the main drivers affecting the soil bacterial community structure in group A. In group B, 1wL was positively correlated with S-SC, TKN, TP, and both 0wL and 2wL were positively correlated with $NH_4^+-N$, S-CAT (Figure S6b). The above results showed that there was a significant correlation between the structure of the rhizosphere soil bacterial community and the physicochemical properties of the soil between the different treatment groups. In addition, it is also confirmed that changes in the bacterial community composition and structure in LDPE-containing microplastic and biomass soils affect the physicochemical properties of the soil.

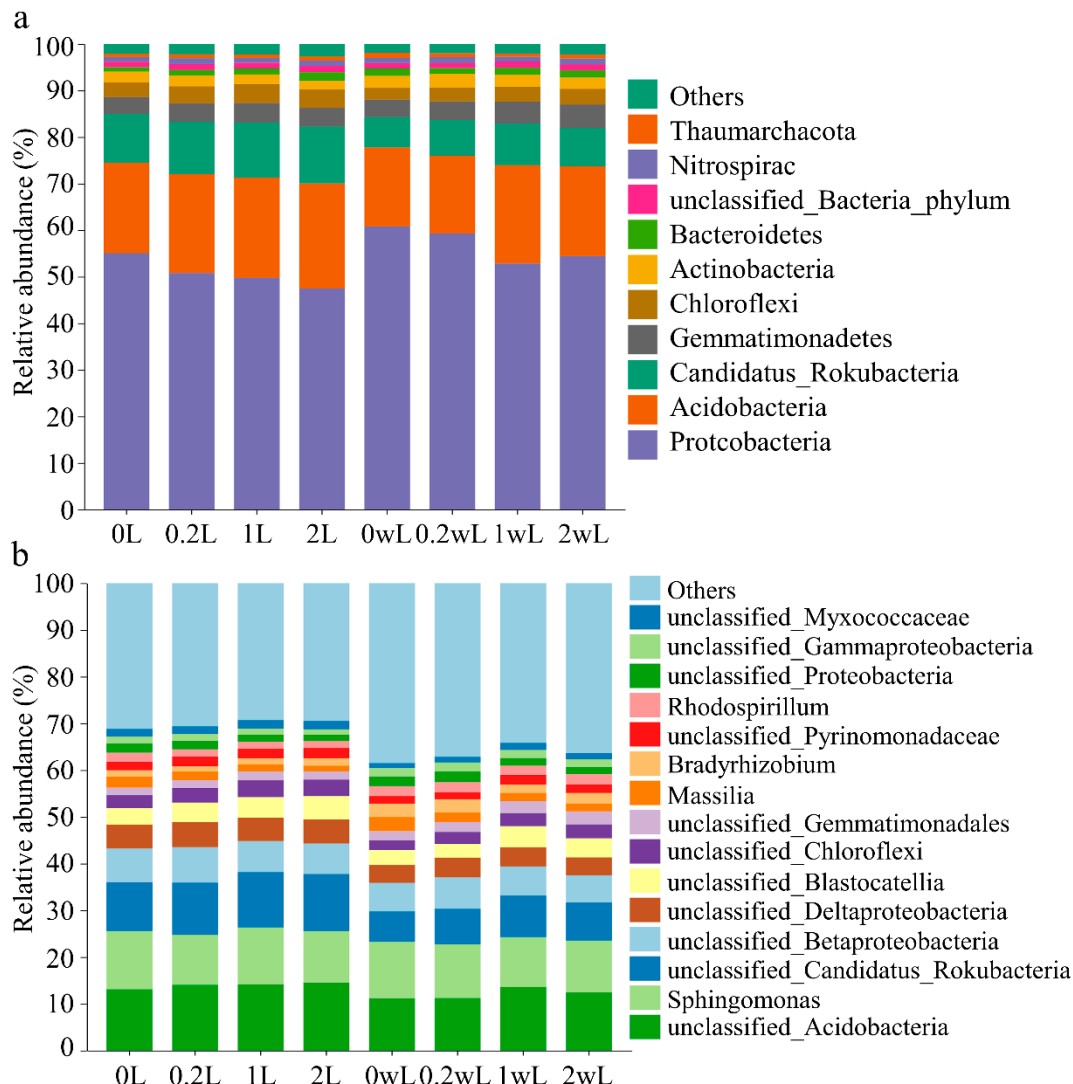

**Figure 4.** Differences in the structure of the rhizosphere soil bacterial community in the different treatment groups. (**a**,**b**) indicate the relative abundance of rhizosphere soil bacteria at the phylum and genus levels in the different treatment groups, respectively.

### 3.6. Effects of LDPE Microplastics and Reed Biomass on Rhizosphere Soil Metabolism

A total of 297 metabolites were identified by GC-MS. The soil metabolites were mainly divided into 12 categories; lipids and lipid-like molecules, organoheterocyclic compounds, and organic acids and derivatives were the major categories of these identifiable compounds, accounting for 31.63%, 20.00%, and 12.09% of the total metabolites, respectively. The proportions of the metabolites in the classifications of benzenoids (10.23%), organic nitrogen compounds (6.51%), and organic oxygen compounds (6.51%) were higher than 5.0% (Figure 5a). As shown in Figure 5b, the 0.2% addition significantly suppressed the abundance of the soil metabolites in the microplastics-only treatment. The abundance of soil metabolites was significantly increased by 0.2% and 1% microplastic addition in group B. Overall, the treatment groups with microplastic exposure and the abundance of soil metabolites showed a trend of increasing and then decreasing with increasing doses of LDPE microplastics, with or without a biomass addition. The quality control (QC) effect was good based on the PCA model analysis, indicating that the data were reliable (Figure S7). The OPLS-DA model analysis showed that the metabolic status of the rhizosphere soil was significantly disturbed in all the different treatment groups. Heatmaps showed the differential metabolites in different treatments for 9 (5 up-regulated, 4 down-

regulated), 15 (11 up-regulated, 4 down-regulated), 6 (1 up-regulated, 5 down-regulated), 13 (3 up-regulated, 10 down-regulated), 9 (5 up-regulated, 4 down-regulated), and 10 (2 up-regulated, 8 down-regulated) (Figure S7a). Among them, both the 0.2% and 1% addition in the microplastic-only treatment significantly suppressed the relative abundance of bisdemethoxycurcumin. However, in group B, both the 0.2% and 1% microplastic additions promoted the relative abundance of PA (20:1(11Z)/15:0), but high doses of LDPE microplastic additions significantly inhibited the relative abundance of PA (20:1(11Z)/15:0) (Figure S8).

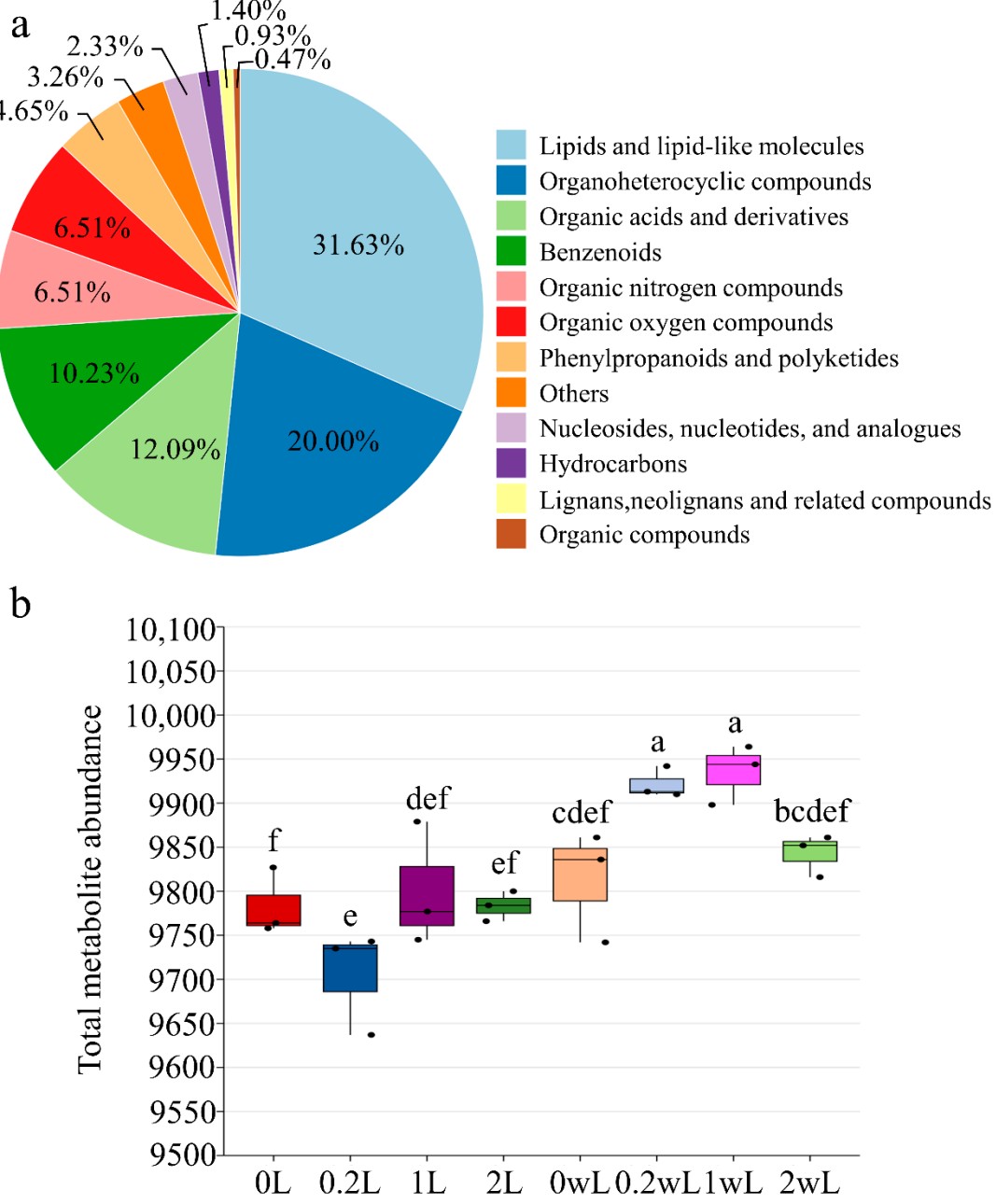

**Figure 5.** Analysis of rhizosphere soil metabolites in each treatment group. (**a**) shows the relative abundance of metabolites in rhizosphere soil. (**b**) shows the difference analysis of the relative abundance of metabolites in rhizosphere soil in each treatment group. Different lowercase (a, b, c, d, e, f) letters indicate significant differences between different treatments ($p < 0.05$, $n = 3$).

### 3.7. Correlation Analysis of Physicochemical Properties and Soil Community Structure

As shown in Figure 6, the bacterial diversity, dominant phyla, and soil metabolism were all significantly influenced by a variety of environmental factors. In group A, the diversity of the soil bacterial community in the reed root was significantly affected by the TOC, $NO_3^--N$, and TP, the dominant bacteria in the reed root were significantly affected by the TOC, TKN, and pH, and the soil metabolites were significantly affected by the S-SC, $NO_2^--N$, $NO_3^--N$, and TP (Figure 6a). In group B, the diversity of the soil bacterial community in the reed root was significantly affected only by $NO_3^--N$, the dominant bacteria in the reed root were significantly affected by the TOC, $NO_2^--N$, TKN, and TP, and the soil metabolites were significantly affected only by the S-SAT (Figure 6b).

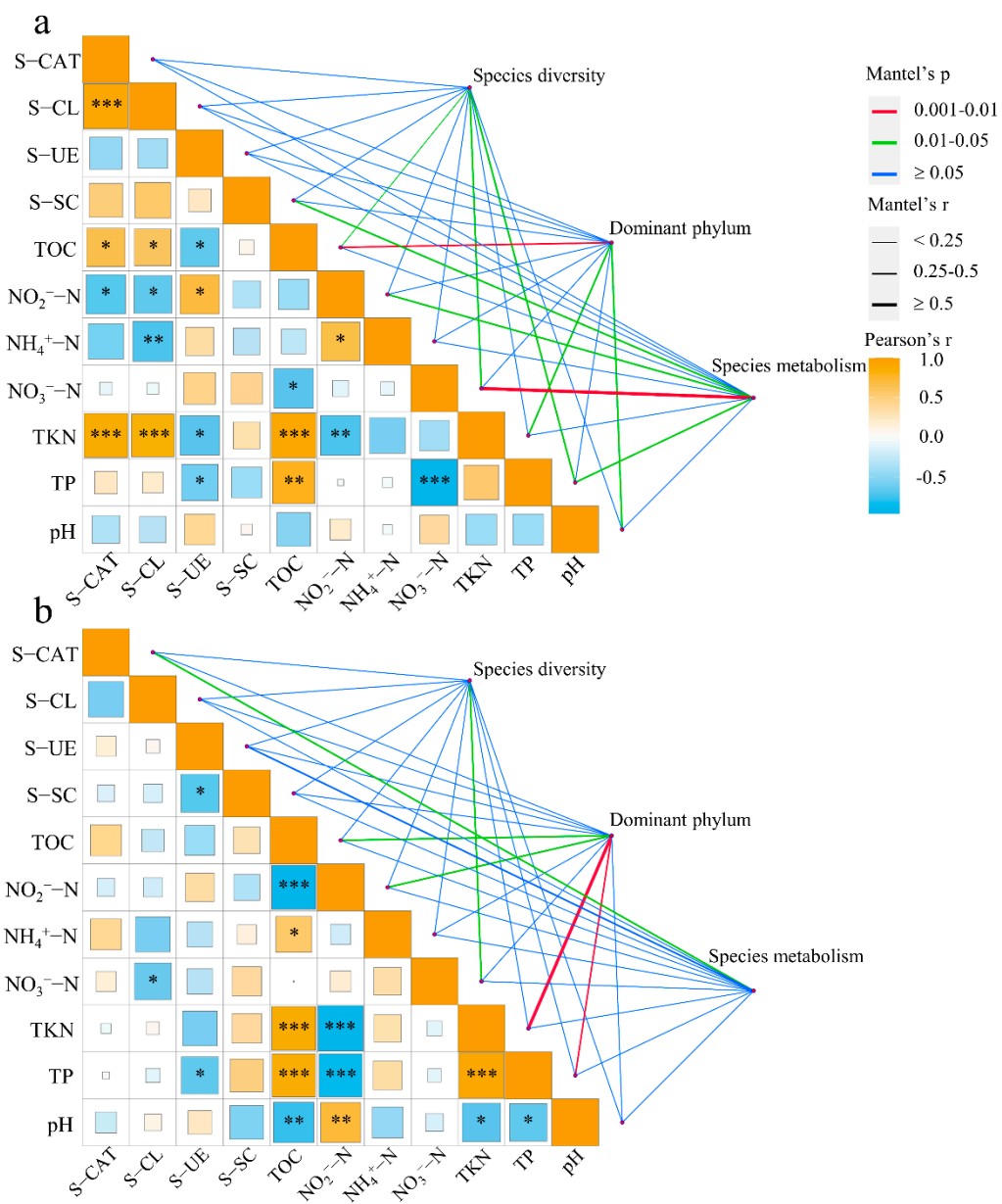

**Figure 6.** Heat map of correlation between rhizosphere soil environmental factors and bacterial community diversity, dominant phyla, and metabolites of bacterial community. (**a**) shows the groups without biomass addition. (**b**) shows the groups with biomass addition. * Significant correlation at $p < 0.05$, ** Significant correlation at $p < 0.01$, and *** Significant correlation at $p < 0.0001$.

## 4. Discussion

MPs retained in soil may interfere with the soil's physical and chemical properties, which has been partially confirmed, and such changes are largely due to factors such as the soil type, microplastic species, concentration, and retention time of the MP residues. The data of this study showed that the physical and chemical parameters of soil fluctuated to some extent with the increase of LDPE microplastic dosage. In group A, the $NH_4^+$-N content of the soil showed a decreasing trend as the microplastic content increased, but none of them were significantly different from the control group. With the addition of the reed biomass, the $NH_4^+$-N content of each treatment group was higher than that of the same microplastic addition level group in group A. This suggests that soil bacteria may be more biased towards degrading readily degradable organic matter and that in experimental group A, high doses of microplastics showed stronger inhibition of nitrogen cycle-associated bacteria, which was moderated by the addition of the biomass [19]. In addition, the $NO_3^+$-N content in groups A and B was significantly higher in the 0.2% spiked level group than in the 2% spiked level group, showing a low promotion and high suppression phenomenon. It may be that a small dose of microplastics increased the porosity of the soil, promoted the circulation of oxygen, and enhanced the nitrification reaction [20]. The addition of high-dose microplastics can increase the porosity of the soil, but due to the high microplastic concentration, the substances that enter the soil after decomposition also increase, which inhibits the nitrification reaction to some extent. Microplastics themselves are composed of a large number of carbon elements, and long-term input will cause soil carbon accumulation, which explains the phenomenon that the TOC content of the soil in this experiment increased with the increase of the microplastic addition [21]. In this study, the content of the TP in the soil increased significantly in the middle and high dosage groups of the microplastics, which was similar to the results of Yin et al. [22]. The higher dosage stimulated the activity of extracellular enzymes related to the phosphorus cycle. Microplastics have been reported to alter the exchange of cations and protons, leading to changes in the pH [23]. However, Wu's [18] 2-month study of different doses of LDPE microplastics added to soil found no significant effect on the pH, either as a time factor or as an additive volume factor. This conclusion is further supported by our data.

A large number of studies have observed the physical/chemical degradation and biodegradation of microplastics in the natural environment through scanning electron microscopy [24]. The SEM results of this study showed that LDPE microplastic particles exposed to soil underwent varying degrees of decomposition. In addition, microplastics have the ability to adsorb xenobiotic chemicals and are, therefore, considered to be carriers of harmful pollutants [25–27]. This study also showed that various elements (such as Mg, Al, Si, K, Ca, Fe, etc.) were adsorbed on the surface of the PE-particles in the soil. It is worth noting that more elements were adsorbed on the surface of microplastics in the biomass treatment. This was due to the addition of the biomass recruiting more degradative bacteria, which roughened the surface of the microplastic and enhanced its adsorption. The ACE index between the different treatment groups also supported these findings. In addition, some reports show that microplastics provide carriers for the attachment of heavy metals and organic pollutants in soil, which increases the harmful effects on the soil structure and microbial activities, and then threatens plant and human health. As the polyethylene particles have a negative surface potential, this leads to reduced anion adsorption due to electrostatic repulsion. In addition, Dong et al. [28] found that the adsorption capacity of microplastics to metal was mainly influenced by the particle size of the microplastics. At present, a lot of research is devoted to synthesizing new materials by photocatalysis to reduce this risk [29,30].

Numerous studies have shown that microplastics can stimulate soil enzyme activity, which in turn affects the cycling of carbon, nitrogen, and phosphorus in the soil [4,31,32]. We observed a differential response of different types of soil enzyme activity to the addition of LDPE microplastics. The addition of the reed could have significantly increased the soil

urease and invertase, which is because the easily degradable biomass could have provided rich nutrient sources for the bacteria, which can enhance the activity and metabolism of soil bacteria, thus significantly increasing the soil urease and invertase. Urease is closely linked to the nitrogen cycle in soil, and in particular by promoting the hydrolysis of nitrogenous organic matter [33]. This study found that the presence of microplastics in group B significantly increased the urease activity in the soil compared to group A. Nitrogen cycle enzyme activity has been reported to decrease with decreasing carbon availability [34]. Therefore, the depletion of carbon in the refractory LDPE microplastics may lead to a reduction in urease activity. Catalase is considered to be an indicator enzyme for aerobic bacteria and is closely associated with aerobic bacterial populations [35]. This study showed that a 2% LDPE microplastic addition significantly increased soil peroxidase activity with or without biomass incorporation. This might be due to the addition of large amounts of microplastics that increase porosity in the soil.

It is well known that in soil ecosystems, microplastics affect the soil bacterial community in two main ways. On the one hand, microplastics change the environment for soil bacteria by altering the physical properties and nutrient conditions of the soil. On the other hand, microplastics stimulate the production of new dominant flora by altering root secretions and bacterial metabolites in the rhizosphere soil. Therefore, we analyzed the diversity and metabolites of rhizosphere soil bacteria in each treatment group. We found that although the Venn diagram showed varying degrees of reduction in the abundance of species-level bacteria in the rhizosphere soil following the addition of LDPE microplastics, there was no significant effect on the $\alpha$ diversity, which was similar to Ya's [36] research. These phenomena can be explained by the fact that soil bacteria are usually resistant to external disturbances [37]. After four months of incubation, Li et al. [38] found that both LDPE and PBAT (polyadipate/butylene terephthalate) microplastics with different additive levels significantly altered the alpha diversity of soil bacteria. Lou et al. [39] showed that the $\alpha$ diversity index of bacteria on the surface of particles in microplastics was significantly lower than that of surrounding soil samples. However, the study by Huang et al. [40] showed that PE microplastics failed to significantly alter the alpha diversity of soil bacteria. This suggests that soil microorganisms are, to some extent, resistant to external disturbances.

Species abundance analysis showed that the predominant phylum in all of the samples was *Proteobacteria*, which is consistent with the results of other studies on soil microplastic contamination [41]. Notably, this study found that the abundance of *Proteobacteria* decreased with the increase of the microplastic content of LDPE in group A. Previous studies have shown that most *Proteobacteria* belong to eutrophic bacteria. With the increase of refractory LDPE microplastics, the nutrients that can be directly utilized by bacteria in the soil decrease, thus reducing their relative abundance. The addition of biomass in group B provided a rich source of carbon for soil bacteria compared to group A, thus significantly increasing the abundance of *Proteobacteria*. Many studies have been conducted to obtain some degrading bacteria by adding microplastics to the soil. These bacteria are mainly concentrated in the phyla *Proteobacteria*, *Bacteroidetes*, and *Firmicutes*, as well as some of their genera, such as *Pseudomonas*, *Bacillus*, and *Acinetobacter* [7,40,42]. A study by Luo et al. [43] showed that *Sphingomonas*, *Bacillus*, and *Gp4* were the dominant core species in agroecosystems with microplastic levels greater than 300 pieces kg$^{-1}$. The results of this study differed from those of previous studies. In this study, the relative abundance of *Candidatus_Rokubacteria* and *Chloroflexi* was significantly increased by both 1% and 2% microplastic addition. In addition, the *unclassified_Blastocatellia_genus*, which belongs to *Acidobacteria*, was significantly enriched in the medium-high microplastic additive group. Although these bacteria showed significant variation with the LDPE microplastic addition, their function in microplastic-contaminated soil ecosystems needs to be further validated to expand the database of possible degrading microorganisms for microbial degradation of soil MPs.

The principal component analysis showed that 1% and 2% LDPE microplastic additions significantly altered the bacterial community structure of the rhizosphere soil. Previous studies have also proved that high concentrations of microplastics can change the β diversity of soil bacteria [38]. In this study, it was found that LDPE microplastics had less effect on the beta diversity of soil bacteria in group B compared to group A. This may be because the addition of exogenous biomass weakened the dose-effect of the soil bacteria on the LDPE microplastics. This study found that the addition of the exogenous biomass weakened the dose-effect of the soil bacteria on the LDPE microplastics. This indicates that the bacterial communities in the soil treated by LDPE microplastics prefer to use external energy and carbon sources [44].

Changes in the composition and content of the soil metabolites may be a direct reflection of the response of the soil microorganisms to exogenous soil contaminants [31,45]. The presence of nano-materials has been reported to alter the distribution of metabolites in the soil and significantly alter metabolic pathways associated with sugars, amino acids, etc. [46]. Organic fertilizer application increases soil amino acid content by influencing amino acid metabolic pathways [47]. In addition, after 60 days of research, Fu et al. [8] found that LDPE microplastics with different molecular weights could significantly change the soil's metabolic state. This study showed that all the treatment groups significantly altered the metabolite profile of the soil after 90 d, including lipids and lipid-like molecules, organic heterocyclic compounds, organic acids and their derivatives, and benzenoid compounds. This confirms some of the preliminary findings that LDPE microplastics can modify metabolites in the rhizosphere soil of reeds. More importantly, in experimental group A, there was a down-regulation of the differential metabolites Harderoporphyrin and LysoPE (18:2(9Z,12Z)/0:0), especially LysoPE (18:2(9Z,12Z)/0:0). In addition, bis-demethoxycurcumin was significantly down-regulated in the 0.2 L and 1 L groups. These metabolites have a certain degree of antioxidant effect, which can slow down the disease risk of plants and bacteria. This further confirmed that the existence of LDPE microplastics aggravated the potential risk to the soil ecosystem. Previous studies have shown that LysoPE (18:2(9Z,12Z)/0:0) belongs to the glycerophospholipid class and is mainly involved in protein recognition and signaling by the cell membrane [48]. In addition, the addition of the reed biomass promoted the relative abundance of N-Acetyl-2, 6-diethylaniline at medium (1%) and high (2%) microplastic additions. The function of these marker metabolites in the soil–plant system remains to be further validated.

## 5. Conclusions

Plastics are widely used materials, and the resulting microplastic pollution is a seriously harmful and expanding global problem, which leads to serious environmental impacts and threatens human health. The purpose of this study was to explore the effects of microplastics and biomass addition on the biodiversity and metabolism of reed rhizosphere soils. The results of this study confirmed that different additive amounts of polyethylene microplastic particles significantly interfered with the physicochemical parameters and enzymatic activity of the reed rhizosphere soil. In addition, after 90 d of soil incubation, the surface of the LDPE pellets underwent varying degrees of biodegradation and adsorbed a variety of elements. Although the α diversity did not show significant differences in the treatment groups, the β diversity showed a clear separation between low and medium-to-high doses, while the addition of biomass weakened the dose-effect generated by the microplastics. The results of the metabolomics study showed that LDPE microplastic particles significantly altered the metabolite profile of reed rhizosphere soil and significantly inhibited the metabolism of glycerophospholipids. This study provides a scientific reference for understanding the effects of LDPE microplastic particles on the rhizosphere soil microenvironment. Based on the current research situation, further research should expand the types of other pollutants that may be loaded into the soil environment in microplastics so as to evaluate the impact of the combination of multiple pollutants on soil ecology.

**Supplementary Materials:** The following supporting information can be downloaded at: https://www.mdpi.com/article/10.3390/w15081505/s1, Figure S1: Thermogravimetric analysis of virgin polyethylene microplastic pellets; Figure S2: Energy dispersive spectrometer of the adsorption characteristics of soil elements on the surface of LDPE microplastic granules. Raw LDPE microplastic pellets (C), The red box indicated the area analyzed by energy dispersive spectrome-ter. The data in the figure represented the percentages of different elements on the surface of LDPE microplastic granules; Figure S3: Venn diagram based on the level of soil bacterial species in the rhizosphere of the different treatment groups; Figure S4: Analysis of the alpha diversity of the rhizosphere soil bacterial community in different treatment groups; Figure S5: The β diversity of rhizosphere soil bacteria in the different treatment groups; Figure S6: a and b indicate the response of rhizosphere soil bacteria to physicochemical parameters in the absence and presence of biomass additions, respectively. Note: catalase (S-CAT), urease (S-UE), cellulase (S-CL), and sucrase (S-SC). In Figure d, "*" indicate significant differences at the confidence levels of 0.05; Figure S7: a shows the distribution of metabolites in different treatment groups through OPLS-DA model analysis. Figure b shows the distribution of differential metabolites in all treatment groups by principal component analysis, in which QC is quality control, and mutual clustering of QC indicates good quality control and reliable data; Figure S8: Differential metabolites in each treatment group, and "*" indicates that there is a significant difference at $p < 0.05$ level; Table S1: Basic chemical properties of the test soils; Table S2: Analysis on the difference of relative abundance of bacterial phylum in rhizosphere soil of each treatment group; Table S3: Analysis on the difference of relative abundance of bacterial genus in rhizosphere soil of each treatment group.

**Author Contributions:** Z.T.: Writing—original draft, Data curation. B.L.: Writing—review and editing, Project administration. W.Z.: Investigation. F.L.: Investigation. J.W.: Visualization. Z.S.: Conceptualization. Y.Z.: Writing—review and editing, Supervision. All authors have read and agreed to the published version of the manuscript.

**Funding:** This work was supported by the project supported by the Science and Technology Research of Henan Province, China (222102320395, 232102321044), the Basic research plan for key scientific research projects of higher education institutions in Henan Province, China (21B610003), the Training Program for Young Key Teachers of higher education institutions in Henan Province, China (2020GGJS218), and the Innovative training program for college students in Henan Province, China (202211765048).

**Data Availability Statement:** Not applicable.

**Conflicts of Interest:** The authors declare no conflict of interest.

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
