# Peer review of "Polyethylene Microplastic Particles Alter the Nature, Bacterial Community and Metabolite Profile of Reed Rhizosphere Soils"

_water, doi:10.3390/w15081505_

Round 1
Reviewer 1 Report
Nanoplastics have been a topic of worldwide concern because of their high concentration and potential ecological risk in the environment. The topic of the manuscript “Polyethylene Microplastic Particles Alter the Nature, Bacterial Community and Metabolite Profile of Reed Rhizosphere Soils” is interesting and thoroughly investigated by the author. However, require certain clarifications/revision for improving the basics of the manuscript.
1. It is recommended to add an graphical abstract, which will help the readers to recognize this study.
2. The abstract section should briefly introduce the research background and significance, clarify the research methods, introduce the main research results, and finally, give the corresponding conclusions. The abstract of this article looks ok, but some descriptions are redundant, and I hope to make a comprehensive modification.
3. The material and method section needs to be re-integrated and checked to avoid repetition and confusion. All experimental protocols should be better explained.
4. The number of biological replicates and technical replicates of this study was not mentioned in the materials and methods.
5. It is advised that the TEM analysis be supplemented to account for changes in the size and shape of microplastic samples under the action of microorganisms. The analysis of TEM results has been recently reported in the literature for reference (https://doi.org/10.1016/j.jclepro.2022.134589;https://doi.org/10.1016/j.jhazmat.2022.129940).
6. Added a Section 3.6 about implications of this work.
7. Discussion: please cited more relevant research to discuss. Here, I can provide some for you: doi.org/10.1016/j.apsoil.2022.104696; doi.org/10.1016/j.jhazmat.2021.126865; doi.org/10.1016/j.envpol.2022.119159
Author Response
请看附件。

Reviewer 2 Report
A good study on a relevant topic. I have the following observations to strengthen the paper:
English form: fairly good in general, please double check the abstract that can be smoothed a bit more. In the Materials section, please avoid imperative form “Remove and sieve to a diameter of 95 2-0.6 mm and store” and replace with “..and then removed and sieved to a diameter..”.
Intro: in the context of the study, it might be worth mentioning that PE can also affect plant growth, see for instance https://doi.org/10.1016/j.jhazmat.2022.129314
Fig. 3: please make it overall easier to read, better quality/larger font in edx spectra, larger sem images, if the final image gets too large consider moving something to a SI file.
Fig. 5-6 same as above, too crowded and small fonts, please enlarge and consider moving something to the SI.
Reviewer 3 Report
The paper is very well structured, with adequate amount of data is well presented and interpreted. The graphical and diagrammatic analyses are thorough. The paper needs to be corrected by following the points:
1. The introduction part can be elaborated by highlighting more on the microplastic-soil-microbe-plant interactions by citing previous studies.
2. Line 46-48: "In addition, microplastics 46 can absorb other pollutants (such as heavy metals and antibiotics) in the soil, which has a far-reaching impact on the ecosystem." Add reference/s in support of the statement. The authors can check the following reference:
Kumar R, Ivy N, Bhattacharya S, Dey A, Sharma P. Coupled effects of microplastics and heavy metals on plants: Uptake, bioaccumulation, and environmental health perspectives. Sci Total Environ. 2022 Aug 25;836:155619. doi: 10.1016/j.scitotenv.2022.155619.
3. Line 195-201: "With the increase of microplastics content, NO3 -N decreased first and then increased, while TP increased first and then decreased with 196 the increase of microplastics content. Compared with the group B, NH4 + -N, NO3 - -N increased initially, followed by a decreased and then increased again with increasing microplastic amount, while TKN and TP increased and then decreased with the increase of microplastics. It was worth noting that the TOC content in reed root soil increased significantly with the increase of microplastics in both group A and group B."
Several increasing and decreasing trends are shown without explaining the possible reasons. The authors should try to explain the possible underlying reasons in details.
4. Please increase the size of the images in figure 3. These images are extremely small, and the main features are not visible clearly.
5. Elaborate the conclusion section by adding the implication of the study and mention of future research perspectives in microplastic effects on soil microbes and consequences.
Round 2
Reviewer 1 Report
It can be accepted.
Reviewer 3 Report
The authors have addressed the comments and made corrections in the manuscript accordingly. The manuscript can be considered for publication.